# Real-World Outcomes in Patients with Metastatic Colorectal Cancer in Spain: The RWD-ACROSS Study

**DOI:** 10.3390/cancers15184603

**Published:** 2023-09-17

**Authors:** Carles Pericay, Ana Fernández Montes, Vicente Alonso Orduña, Ismael Macias Declara, Elena Asensio Martínez, Nuria Rodríguez Salas, Esperanza Torres, Diego Cacho Lavín, Rosa María Rodríguez Alonso, Esther Falcó, Joan Carles Oliva, Lluis Cirera

**Affiliations:** 1Servicio de Oncología Médica, Hospital Universitario Mútua Terrassa, Plaça del Doctor Robert, 5., 08221 Terrassa, Spain; lcireran@hotmail.com; 2Servicio de Oncología Médica, Complexo Hospitalario Universitario de Ourense, Calle Ramón Puga Noguerol, 54., 32005 Ourense, Spain; afm1003@hotmail.com; 3Servicio de Oncología Médica, Hospital Universitario Miguel Servet, Instituto de Investigacion Sanitaria de Aragon, Paseo Isabel la Católica, 1-3., 50009 Zaragoza, Spain; valonsoonco@gmail.com; 4Institut d’Investigació I Innovació I3PT, Fundació Parc Taulí, Plaça Taulí, 1., 08208 Sabadell, Spain; imacias@tauli.cat (I.M.D.); joanc.oliva@gmail.com (J.C.O.); 5Servicio de Oncología Médica, Hospital General Universitario de Elche, Carrer Almazara, 11., 03203 Elche, Spain; helenasensio@yahoo.es; 6Servicio de Oncología Médica, Hospital Universitario La Paz, Paseo de la Castellana, 261., 28046 Madrid, Spain; nuria.rodriguez@salud.madrid.org; 7UGC Intercentros de Oncología Médica, Hospitales Universitarios Regional y Virgen de la Victoria and Instituto de Investigación Biomédica de Málaga (IBIMA), Campus de Teatinos, S/N, 29010 Málaga, Spain; esp_torres2001@yahoo.es; 8Servicio de Oncología Médica, Hospital Universitario Marqués de Valdecilla, 39008 Santander, Spain; juandiego.cacho@scsalud.es; 9Servicio de Oncología Médica, Hospital Universitario Reina Sofía-Córdoba, Avenida Menéndez Pidal s/n., 14004 Córdoba, Spain; rosarodriguezalonso@gmail.com; 10Servicio de Oncología Médica, Hospital de Son Llàtzer, Carretera de Manacor, 07198 Palma de-Mallorca, Spain; efalco@hsll.es

**Keywords:** colon cancer, colonic neoplasms, continuum of care, prognostic factors, real-world data, Spain

## Abstract

**Simple Summary:**

Metastatic colorectal cancer is common and complicated to treat. The first choice of treatment is guided by the genetics of the patients (*RAS* mutation status) and the location of their primary tumor. However, real-world information on what treatments are actually being used and the survival benefits of these patient-guided treatments in Spain is lacking, so we aimed to determine this. We found that, in general, treatment choice was guided by primary tumor location and mutation status, with mutation status being the more influential characteristic. In addition, non-mutated *RAS* or left-sided tumors were predictive of longer survival times. We hope the results of this study may serve as a useful benchmark for future studies and assist practicing oncologists in selecting appropriate treatments for their patients.

**Abstract:**

The retrospective, observational RWD-ACROSS study analyzed disease characteristics, systemic treatment, and survival in patients with metastatic colorectal cancer (mCRC) in Spain. In total, 2002 patients were enrolled (mean age 65.3 years; 62.7% male). Overall median overall survival (OS) was 26.72 months, and was longer in patients with left-sided tumors (28.85 vs. 21.04 months (right-sided tumors); *p <* 0.0001) and in patients receiving first-line anti-epidermal growth factor receptor (EGFR) treatment (31.21 vs. 26.75 (anti-vascular endothelial growth factor (VEGF) treatment) and 24.45 months (chemotherapy); *p =* 0.002). Overall median progression-free survival (PFS) was 10.72 months and was longer in patients with left-sided tumors (11.24 vs. 9.31 months (right-sided tumors); *p <* 0.0001), and in patients receiving either first-line anti-EGFR or anti-VEGF (12.13 and 12.00 vs. 8.98 months (chemotherapy); *p <* 0.001). PFS was longer with anti-VEGF treatment in patients with right-sided tumors and wild-type *RAS* (11.24 vs. 8.78 (anti-EGFR) and 7.83 months (chemotherapy); *p =* 0.025). Both anti-EGFR and anti-VEGF produced longer PFS in patients with left-sided tumors and wild-type *RAS* than chemotherapy alone (12.39 and 13.14 vs. 9.83 months; *p =* 0.011). In patients with left-sided tumors and mutant *RAS*, anti-VEGF produced a longer PFS than chemotherapy alone (12.36 vs. 9.34 months; *p =* 0.001). In Spain, wild-type *RAS* or left-sided mCRC tumors are predictive of longer survival times.

## 1. Introduction

Colorectal cancer (CRC) is the most common type of cancer in Spain, with 42,721 new CRC diagnoses estimated for 2023 [1]. Approximately 25% of patients with CRC have metastatic CRC (mCRC) at diagnosis and 20–30% of patients diagnosed with stages I–III develop metastases during follow-up. Overall, approximately 45% of patients with CRC will be treated for metastatic disease at some point in their lives [2].

Clinical trials in the late 1990s established the central role of oxaliplatin- and irinotecan-based regimens in the first-line treatment of mCRC [3,4,5,6]. Subsequent trials demonstrated the ability of adjunctive targeted therapies, such as bevacizumab, cetuximab, and panitumumab, to further improve survival outcomes [7,8,9,10,11]. Since 2010, new cytotoxic agents (e.g., trifluridine/tipiracil (TAS-102) [12]) and targeted therapies (e.g., aflibercept [13], ramucirumab [14], and regorafenib [15]) have expanded the range of available treatment options, particularly in second and later lines of therapy. Additionally, pembrolizumab, an antibody that blocks programmed cell death protein 1 (PD-1) on the surface of lymphocytes, has emerged as a highly effective option for patients with microsatellite instability (MSI)-high tumors with mismatch-repair deficiency [16].

Since 2008, evidence has accumulated showing that mutations in the *KRAS* (exons 1–4), *NRAS* (exons 1–4), and *BRAF* genes are predictive of poor response to anti-epidermal growth factor receptor (EGFR) therapy in patients with mCRC [17]. In patients with wild-type *RAS*, four studies have directly compared anti-EGFR with anti-vascular endothelial growth factor (VEGF) therapy (both in combination with chemotherapy) as first-line therapy for mCRC (PEAK, FIRE-3, and CALGB/SWOG 80405, as reviewed by Mitchell and Sanoff [18], and the PARADIGM study [19]). A meta-analysis of the first three of these trials found both types of treatments to be associated with prolonged overall survival (OS) and progression-free survival (PFS) and higher overall response rates (ORR) in patients with left-sided versus right-sided primary tumors [20]. Furthermore, the effect of the tumor side on treatment outcomes was larger for anti-EGFR-based treatment than for anti-VEGF-based treatment [20]. In particular, PARADIGM showed a clear efficacy advantage for anti-EGFR in left-sided tumors [19].

As a result of these studies, the choice of systemic first-line treatment for patients with mCRC is now guided by *RAS* mutation status and primary tumor location [21,22]: cetuximab and panitumumab are preferred for left-sided, wild-type *RAS* tumors, whereas bevacizumab is preferred for right-sided tumors and *RAS*-mutated tumors regardless of their location. However, despite the prognostic and predictive importance of *RAS* mutation status and primary tumor location, treatment is sometimes initiated without this information.

Indeed, real-world data on mCRC are notably lacking in Spain. Population-based registries provide high-level data on mCRC incidence and mortality; however, no national databases provide detailed insights into patient and disease characteristics, treatment choices, and outcomes in oncology. Thus, our knowledge of primary tumor location, *RAS* mutation status, MSI status, and the type and duration of targeted therapy administered (overall and by the line of therapy) is incomplete. Additionally, we do not know the effectiveness of systemic treatments for mCRC in real-world clinical practice in Spain, nor do we know the optimal treatment sequence to maximize quality-of-life adjusted survival for different patient groups.

To address this, we undertook a retrospective analysis to gather and analyze comprehensive data on disease characteristics, systemic treatment, and survival in patients with mCRC across ten hospitals in Spain.

## 2. Materials and Methods

RWD-ACROSS was a retrospective, observational study of patients with mCRC conducted across ten specialist oncology centers. Participating centers were selected from eight of the 17 autonomous communities in Spain, covering 71.4% of the population. Collaborating centers were selected in this way to reduce the effect of potential biases resulting from differences in pharmacological approvals between regional and local health authorities, and to lessen the impact of variations in practice between centers.

### 2.1. Inclusion and Exclusion Criteria

Each collaborating center contributed all patients in their registry, up to a maximum of 250 patients, who received a diagnosis of mCRC during the study period (January 2011–December 2015) and met the inclusion criteria. Patients had to have histologically confirmed mCRC, be aged ≥18 years at the time of diagnosis, and have received ≥1 cycle of chemotherapy for metastatic disease. Previous resection of a primary tumor and subsequent adjuvant treatment was permitted. Additionally, patients needed to have a complete set of medical records for ≥1 year, including and following primary diagnosis of mCRC.

Patients were excluded if they had a synchronous or metachronous malignant tumor diagnosed during the study period, which, in the investigator’s opinion, could have affected their prognosis. Additional exclusion criteria included the administration of any mCRC treatment or resection of metastasis with R0 before January 2011, with or without chemotherapy. Previous adjuvant treatment following primary tumor resection was permitted for patients with metachronous metastatic disease.

### 2.2. Outcomes

The main objective of the study was to determine OS and PFS for the entire study cohort, and for patients with known prognostic or predictive factors. In the latter analyses, OS and PFS were estimated according to primary tumor location (right-sided vs. left-sided), *RAS* mutation status (wild-type vs. mutated), therapy type (chemotherapy alone or in combination with an anti-VEGR/anti-EGFR agent), line of therapy, and first-/second-line treatment sequence. Additional objectives were to characterize the patient population in terms of demographic and clinical variables, and to analyze treatment by line of therapy.

All patient data were obtained from medical records and included age, sex, dates of primary tumor and metastatic disease diagnosis, and date of death or last follow-up. Tumor characteristics included location (right-sided or left-sided), the number and location of metastases, *RAS* and *BRAF* mutation status, and MSI status. ‘Right-sided’ was defined as primary disease in the caecum, ascending colon, and transverse colon, and ‘left-sided’ included tumors from the splenic flexure to the rectum, inclusive. Data on treatment, including resection of the primary tumor and metastases, were also collected. Where applicable, details of administered chemotherapy and targeted therapies, alongside their dates of administration, were extracted for each of the first three lines of treatment; the initiation dates of any fourth-line treatments were also extracted.

The protocol was approved by the Ethics Committee for Drug Research at each participating hospital prior to study commencement (approval number ACR-QUI-2018-01).

### 2.3. Statistical Analyses

Patients were included in the analysis set if: (i) they met the inclusion criteria; (ii) the date of mCRC diagnosis was known; and (iii) the patient’s status at the date of last contact was known.

All continuous variables were summarized using the following descriptive statistics: number (n; based on the number of recorded values for each parameter), mean, standard deviation (SD), 95% confidence interval (95% CI), median, interquartile range (IQR), maximum and minimum. The number and percentage of observed values (with the denominator being based on the number of recorded values for each parameter) were reported for all categorical variables.

OS was defined as the time from mCRC diagnosis to the date of death (any cause); survival curves were plotted using the Kaplan–Meier method. Patients without an event (e.g., due to loss to follow-up) were censored at the date they were last known to be alive. PFS was also summarized using the Kaplan–Meier method. First-line PFS was measured from the start of first-line treatment to either death (any cause) or the start of second-line treatment. Patients without an event were censored at the date they were last known to be alive and not receiving second-line treatment. Similarly, second- and third-line PFS was measured from the start of second- or third-line treatment, respectively, to death (any cause) or the initiation of third- or fourth-line treatment, respectively. Patients without an event were censored at the date they were last known to be alive and not receiving any subsequent-line treatment.

Univariate analysis was performed to identify factors predictive of OS or PFS. The variables included age (≤70 vs. >70 years), sex, clinical characteristics (primary tumor location; *RAS* mutation status), number of metastatic sites at diagnosis, and treatment (surgical resection of the primary tumor or metastases; inclusion of biologic therapy in the first-line regimen). The log-rank test was used. Cox regression was used in subsequent multivariate analysis to identify interactions between variables that were significantly associated with OS or PFS. Backward selection was used to identify factors with independent prognostic value. Odds ratios (ORs) and corresponding 95% CIs were calculated.

All statistical analyses were performed using SAS^®^ version 9.4 (SAS, Cary, NC, USA).

## 3. Results

### 3.1. Patient and Disease Characteristics

A total of 2024 patients were identified during the screening phase. Of these, 2002 patients received ≥1 line of chemotherapy for mCRC during the study period (Table 1 and CONSORT [Consolidated Standards of Reporting Trials] flow diagram in Appendix A). As several hospitals included fewer than the stipulated maximum of 250 patients, a single hospital was allowed to include additional patients (n = 300).

Patients were primarily male (n = 1256; 62.7%) and, at the time of first-line therapy, had a mean age of 65.3 years. Most patients had exclusively left-sided primary tumors (n = 1444; 72.1%), almost three quarters of which (n = 1074; 74.4%) were located in the sigmoid colon or rectum (see Appendix A). Although fewer patients had exclusively right-sided primary tumors, the caecum was the third most common location overall, accounting for 10.0% of all primary and 37.5% of all right-sided primary tumors. *RAS* and *BRAF* mutation statuses were known in 1675 patients (83.7%) and unknown in 327 patients (16.3%). Among the former group, 821 patients (49.0%) had wild-type *RAS* and *BRAF*, 828 (49.4%) had mutated *RAS*, and 26 (1.6%) had mutated *BRAF*.

More than half of the patients (n = 1185/2002; 59.2%) had a single metastatic site, most frequently in the liver (n = 754/2002; 37.7%), lungs (n = 164/2002; 8.2%), or peritoneum (n = 152/2002; 7.6%). In the overall study population (n = 2002; i.e., patients with single or multiple metastatic sites), the liver was also the most frequently involved organ (n = 1404; 70.1%; Table 1).

### 3.2. Treatment

#### 3.2.1. Overall

Systemic treatment for mCRC by line of therapy is shown in Table 2; all patients received chemotherapy as first-line therapy. Patients who received chemotherapy as second- and third-line treatment excluded those who died on first-line treatment and those who underwent surgery for metastasis removal. The proportion of patients who also received a targeted agent was approximately 55% in each line (56.4% in the first and second line, and 54.5% in the third line).

First-line chemotherapy was with an oxaliplatin-based regimen (e.g., FOLFOX, CAPOX) in 65.7% of patients and with an irinotecan-based regimen (e.g., FOLFIRI) in 20.1% of patients. Additionally, a small percentage of patients received first-line FOLFIRINOX (n = 41; 2.0%). The median duration of first-line chemotherapy was 5.47 months overall, 5.27 months for oxaliplatin-based therapy and 6.19 months for irinotecan-based therapy.

In the second line, most patients received an irinotecan-based regimen (n = 824; 60.1%) whilst, in the third line, 42.8% of patients received an irinotecan-containing regimen, 24.0% received an oxaliplatin-containing regimen, and 26.4% received fluoropyrimidine alone.

Among those receiving targeted therapy, the proportions who received an anti-VEGF agent (i.e., bevacizumab or aflibercept) as a part of first-, second-, and third-line treatment were 62.7%, 68.4%, and 53.0%, respectively (Table 2). Corresponding percentages for anti-EGFR therapy (i.e., cetuximab or panitumumab) were 36.8%, 30.5%, and 39.7%, respectively. Regorafenib was used in 0.5%, 1.0%, and 7.1% of patients in the first, second, and third line, respectively.

Bevacizumab and cetuximab were the most used anti-VEGF and anti-EGFR agents in all lines of therapy, respectively. Aflibercept was mostly used as a second-line treatment, while panitumumab was used in approximately 10% of patients in all three treatment lines (Table 2).

The median duration of treatment with a targeted agent was 5.90 months. Among individual agents, median treatment duration was longest for bevacizumab (6.19 vs. 5.83 months for cetuximab, 5.50 months for aflibercept, 4.83 months for panitumumab, and 3.98 months for regorafenib).

#### 3.2.2. By Tumor Location and RAS Mutation Status

The data in this section refers only to patients with a known *RAS* mutation status (Table 1). Anti-EGFR agents were infrequently used in patients with mutant *RAS* (in <5% of patients as first line, and <2.5% of patients as second or third line), regardless of primary tumor location (Table 2).

##### Right-Sided Disease

Among patients with a right-sided primary tumor and wild-type *RAS*, anti-EGFR was the most frequently administered targeted therapy (>35% in all three treatment lines; Table 2). In those with right-sided disease and mutated *RAS*, anti-VEGF agents were administered to approximately half of the patients in first and second treatment lines, and to 35.9% in the third line. At each line of therapy, patients with right-sided tumors and *RAS*-mutated disease were more likely to receive chemotherapy only (than those with right-side tumors and wild-type *RAS*). Indeed, chemotherapy alone was the dominant third-line therapeutic strategy in this patient population.

##### Left-Sided Disease

In patients with a left-sided primary tumor and wild-type *RAS*, there was a preference for anti-EGFR therapy regardless of treatment line (Table 2). In this setting, anti-VEGF therapy was more likely to be used in second-line treatment. Approximately 30% of patients in all three lines received chemotherapy without a targeted agent.

Among those with left-sided disease and mutated *RAS*, anti-VEGF agents were used as first-line therapy in 52.3% of patients, as second-line therapy in 48.7%, and as third-line therapy in 43.1%. Conversely, the use of chemotherapy alone increased from 44.5% in the first line to 49.5% in the second and 55.1% in the third line.

### 3.3. Survival

#### 3.3.1. Overall Survival

In the overall study cohort, median OS following first-line treatment was 26.72 months (95% CI: 25.37–28.07; Table 3, and Appendix A). Median OS was significantly lower in patients with right-sided primary tumors than in those with left-sided tumors (21.04 vs. 28.85 months, respectively; *p* < 0.0001). When analyzed by *RAS* mutation status, median OS was 29.04 months (95% CI: 26.98–31.11) in patients with wild-type *RAS* and 27.54 months (95% CI: 25.57–29.50) in patients with mutated *RAS*.

OS according to first-line treatment, primary tumor location, and *RAS* mutation status is shown in Table 4. Patients who received first-line treatment with an anti-EGFR agent had a median OS of 31.21 months (95% CI: 27.78–34.63), compared with 26.75 months (95% CI: 24.67–28.83) in those who received an anti-VEGF agent and 24.45 months (95% CI: 22.51–26.40) in those who received chemotherapy alone (*p* = 0.002 for group comparisons).

Choice of first-line therapy did not significantly affect median OS in patients with right-sided disease; the median OS was 22.13, 21.63, and 19.37 months with anti-VEGF, anti-EGFR, and chemotherapy alone, respectively (*p =* 0.526). In patients with left-sided disease, patients who received first-line treatment with an anti-EGFR agent had a median OS of 34.29 months (95% CI: 31.04–37.54), compared with 28.23 months (95% CI: 25.68–30.77) in those who received an anti-VEGF agent and 26.59 months (95% CI: 24.23–28.94) in those who received chemotherapy alone (*p* = 0.003 for group comparisons).

Median OS was not significantly affected by choice of first-line therapy in patients with either wild-type or mutated *RAS*. In patients with wild-type *RAS*, median OS was 26.62, 30.06, and 29.14 months with anti-VEGF, anti-EGFR, and chemotherapy alone, respectively (*p* = 0.963). In patients with mutant *RAS*, median OS was 28.85, 32.19, and 25.34 months, respectively (*p* = 0.140).

Choice of first-line therapy did not significantly affect median OS in patients with right-sided disease, and either wild-type or mutated *RAS*. In patients with a right-sided tumor and wild-type *RAS*, median OS was 18.62, 19.31, and 24.62 months with anti-VEGF, anti-EGFR, and chemotherapy alone, respectively (*p* = 0.201). In patients with mutant *RAS*, there was no significant difference in OS between those who received anti-VEGF therapy or chemotherapy alone (23.47 vs. 23.86 months; *p* = 0.907).

In patients with left-sided disease and wild-type *RAS*, median OS duration was 28.06, 33.86, and 30.26 months in patients who received first-line anti-VEGF therapy, anti-EGFR therapy, or chemotherapy only, respectively (*p* = 0.583). In those with mutant *RAS*, there was no significant difference in OS duration between patients who received anti-VEGF or chemotherapy only (*p* = 0.182).

Median OS duration following second- and third-line treatment was 14.03 months (95% CI: 13.08–14.98; Appendix A) and 10.65 months (95% CI: 9.67–11.63; Appendix A), respectively. Patients (n = 1428) who received second-line treatment had a median OS, from initiation of first-line treatment, of 29.21 months (95% CI: 27.89–29.52). Those who received third-line treatment (n = 725) had a median OS, from initiation of first-line treatment, of 35.14 months (95% CI: 33.42–36.87).

Median duration of OS according to first- and second-line treatment sequence is shown in Appendix A. In this analysis, in general, median OS was longest in patients who received a targeted (biologic) therapy in both treatment lines and shortest in patients who received only chemotherapy in either line.

Among patients who received biologic therapy in both lines, a first-line anti-EGFR agent with a second-line anti-VEGF agent was associated with a longer median OS compared with the reverse (31.93 vs. 25.41 months; Appendix A). This pattern was also observed in patients with wild-type *RAS* regardless of their primary tumor location.

#### 3.3.2. Progression-Free Survival

Overall median PFS after first-line therapy was 10.72 months (95% CI: 10.24–11.19; Appendix A). Median PFS in patients with wild-type and mutated *RAS* was 11.24 months (95% CI: 10.53–11.95) and 10.78 months (95% CI: 10.04–11.53), respectively (Table 3). Median PFS was significantly longer in patients with a left-sided primary tumor compared with a right-sided primary tumor (11.24 vs. 9.31 months; *p* < 0.0001).

Median PFS was approximately 12 months in patients who received an anti-EGFR or an anti-VEGF agent as part of first-line therapy, and lower (8.98 months) in patients who received chemotherapy only (*p* < 0.001; Table 5).

In patients with right-sided tumors, first-line anti-VEGF treatment was associated with a significantly longer median PFS than anti-EGFR therapy or chemotherapy alone (11.44 vs. 10.75 or 7.77 months, respectively; *p =* 0.002). In contrast, in patients with left-sided tumors, first-line anti-EGFR treatment was associated with a significantly longer median PFS than either anti-VEGF therapy or chemotherapy alone (13.21 vs. 12.39 or 9.50 months; *p <* 0.0001).

Choice of first-line treatment had no effect on PFS duration in patients with wild-type *RAS*; however, in patients with mutated *RAS*, median PFS with first-line treatment with an anti-VEGF agent was 12.36 months (95% CI: 11.33–13.38), compared with 12.03 months (95% CI: 8.34–15.72) in those who received an anti-EGFR agent and 8.98 months (95% CI: 8.37–9.59) in those who received chemotherapy alone (*p* < 0.0001 for group comparisons).

First-line anti-VEGF treatment was associated with a significantly longer median PFS than either anti-EGFR therapy or chemotherapy alone in patients with right-sided disease and wild-type *RAS* (*p* = 0.025). In patients with right-sided disease and mutated *RAS*, anti-VEGF was associated with a longer median PFS than chemotherapy only but the difference was not statistically significant (*p* = 0.056). In patients with left-sided disease and wild-type *RAS*, there was little difference in median PFS between anti-VEGF and anti-EGFR therapies. Both were associated with a longer median PFS than with chemotherapy alone (12.39 and 13.14 months vs. 9.83 months, respectively; *p* = 0.011). Anti-VEGF therapy was associated with a significantly longer median PFS than chemotherapy alone in those with left-sided disease and mutated *RAS* (12.36 vs. 9.34 months, respectively; *p* = 0.001).

Median PFS times for second- and third-line treatment were 7.63 months (95% CI: 7.25–8.02; Appendix A) and 6.16 months (95% CI: 5.65–6.67; Appendix A), respectively. Further analysis of PFS following second- and third-line treatment is shown in Table 5.

### 3.4. Univariate and Multivariate Analyses

Results of the univariate and multivariate analyses are shown in Figure 1 (OS) and Figure 2 (PFS). In the univariate analysis, a significant association was found for all variables analyzed for OS (Figure 1a) and PFS, except for sex and *RAS* mutation status (Figure 2a). Among factors with a significant association, metastatic surgery was most strongly associated with a positive outcome, closely followed in prognostic importance by resection of the primary tumor.

In the multivariate analysis of OS (Figure 1b), no significant difference in outcome was found between men and women. The main interactions between variables were for sex and age (better OS for women aged ≤70 years), *RAS* mutation status and primary tumor location (better OS in patients with left-sided, *RAS* wild-type tumors), and *RAS* mutation status and number of metastatic sites at diagnosis (better OS in patients with *RAS* wild-type disease and a single metastatic site).

In the multivariate analysis of PFS (Figure 2b), age and *RAS* mutation status were not independently prognostic. The interactions that were significant for PFS were sex and age (i.e., better PFS in women aged ≤70 years), age and metastatic surgery (i.e., better PFS in patients aged ≤70 years who had undergone metastatic resection), and *RAS* mutation status and primary tumor location (i.e., better PFS in patients with a left-sided primary tumor and wild-type *RAS*).

## 4. Discussion

In this paper, we present detailed findings of RWD-ACROSS, the first multicenter study of real-world survival outcomes following first-, second-, and third-line systemic treatment for mCRC in Spain. We analyzed data from over 2000 patients diagnosed with mCRC between 2011 and 2015. The male:female ratio was approximately 3:2 and the mean age was 65.3 years. There was a predominance of left-sided tumors (approximately 3:1) and an equal balance between *RAS* wild-type and mutated tumors. Most patients had a single metastatic site, mainly in the liver; nearly two-thirds had undergone resection of the primary tumor, whereas only one in four had received a metastasectomy. Median OS and PFS were 26.72 and 10.72 months, respectively. As expected, median OS times were longer in patients with either left-sided or wild-type *RAS* tumors.

As per the inclusion criteria, all patients received fluoropyrimidine-based chemotherapy. Since 2004, when Tournigand and colleagues published the seminal GERCOR study results [23], patients have been treated with oxaliplatin- or irinotecan-based chemotherapy as first-line treatment, and for second-line treatment switched to the drug not administered in the first line, as illustrated in the following three studies. A US study of patients with mCRC treated between 2004 and 2011 found oxaliplatin was administered in the first line to 60.7% of patients and irinotecan to 11.2% of patients [24]. A Spanish study of patients with mCRC treated between 2016 and 2017 reported that 66.2% of patients received oxaliplatin-based chemotherapy and 18.7% received irinotecan-based chemotherapy as first-line treatment, followed respectively by 21.5% and 48.9% as second-line treatment [25]. Finally, the randomized CALGB 80,405 study, where the choice of first-line chemotherapy was not prespecified, found that 73.4% of patients received oxaliplatin-based chemotherapy and 26.6% received irinotecan-based chemotherapy as first-line treatment [26]. In our study, 65.7% of patients received oxaliplatin-based therapy and 20.1% received irinotecan-based therapy in the first line, and 27.3% and 60.1% in the second line, respectively, indicating our results are in line with standard practice in Western countries.

Approximately 43.0% of patients in our study did not receive a targeted agent in either the first or second treatment line. The proportions of patients receiving biologic drugs (the anti-VEGF bevacizumab, or anti-EGFR agents cetuximab and panitumumab) as first-line treatment of mCRC have varied subsequent to approval of the first biologic in 2004, in part due to restrictions introduced in 2007 regarding the use of anti-EGFR agents in patients with mutant *RAS* and due to the outcomes of comparative studies of these biologics. In the above-mentioned 2004–2011 US study, biologics were added to the first-line regimen in 55% of patients (51% anti-VEGF and 4% anti-EGFR) [24]. When the analysis was limited to patients treated from 2009 onwards, the percentage of patients treated with an anti-EGFR agent rose to 34%. Similarly, in the abovementioned 2016–2017 Spanish study, biologics were added to chemotherapy as first-line treatment for 61.4% of patients (mainly anti-VEGF (38.8% of patients) and anti-EGFR agents (22%)) [25]. Another European study, by Kafatos and colleagues, of first-line treatment in 2018 of patients with mCRC for whom *RAS* and *BRAF* status was known, reported that 81.0% of patients with wild-type *RAS* tumors received a biologic agent (anti-EGFR + chemotherapy 62.6% of patients; anti-VEGF + chemotherapy 18.4% of patients), whereas 19.0% received chemotherapy alone [27]. A biologic agent was received by 61.9% of patients with mutated *RAS* tumors [27]. In addition to intercountry variation, Kafatos and colleagues’ study reflects prescribing decisions made in 2018, several years after our RWD-ACROSS study was conducted. In our study, 56.4% of patients received a biologic agent (anti-VEGF 62.1% or anti-EGFR 36.8%) as first-line treatment. Given that our study included patients diagnosed between 2011 and 2015, our results reflect an increase in the use of biologics in the first line of treatment. However, our results suggest we are still far from offering biologic treatment to 100% of patients as recommended by European clinical guidelines [2]. Although we did not specifically investigate the reasons for initial and subsequent treatment choices, our data raise the possibility that biologic agents may have been under-utilized in Spain during the study period. It should be noted, however, that our study population showed a broad range of clinical characteristics, and included patients with Eastern Cooperative Oncology Group performance status (ECOG-PS) scores of 2–3 who were ineligible for biologic combination therapy and received only fluoropyrimidine monotherapy, which may help explain our lower-than-expected use of biologics.

Anti-VEGF agents were used more frequently than anti-EGFR agents, regardless of treatment line. However, anti-EGFR use was more common in patients with wild-type *RAS* but rare in patients with mutated *RAS*. Because wild-type and mutant *RAS* occurred with approximately equal frequency, the net effect was greater use of anti-VEGF agents overall. The most commonly used anti-VEGF and anti-EGFR agents were bevacizumab and cetuximab, respectively, although variations were seen in the use of targeted agents across treatment lines. Bevacizumab use decreased at each successive treatment line, while aflibercept was most commonly used in the second line and regorafenib was almost exclusively used in the third line.

Subsequent to 2017 when the aforementioned meta-analysis by Arnold and colleagues was published [20], the results of which were later confirmed by the PARADIGM study [19], patients with a wild-type *RAS* mCRC tumor have generally been treated with a combination of chemotherapy and an anti-EGFR agent. Analysis of the Kafatos et al. study, which was conducted prior to the PARADIGM study, shows that the use of anti-EGFR treatment for this population group did not exceed 63% [27]. In our study 45% of patients with a primary tumor (in the left colon) that was wild-type for *RAS* received anti-EGFR combination therapy; this rate probably reflects our patient enrolment period (up to 2015), which was 1 year prior to publication of the Arnold et al. meta-analysis and 7 years prior to the PARADIGM study. Our results showed that primary tumor location had less influence than *RAS* mutation status on the choice of targeted therapy. Regardless of tumor location, approximately half of the patients with mutated *RAS* received chemotherapy only; the remaining half received chemotherapy plus anti-VEGF in the first and second treatment lines. In the third line, however, chemotherapy only was the dominant strategy in *RAS*-mutated patients irrespective of tumor-sidedness.

Meta-analyses of randomized controlled trials have shown that biologic agents significantly improve survival outcomes when added to standard chemotherapy regimens in mCRC [28,29]. Consistent with this, we found that median OS and PFS were significantly longer in patients who received first-line treatment with a biologic agent than in those who received chemotherapy only. First-line treatment with anti-EGFR therapy was associated with a median OS of 31.21 months and median PFS of 12.00 months, while corresponding values for first-line anti-VEGF therapy were 26.75 and 12.13 months, respectively. In contrast, patients who received chemotherapy only had a median OS of 24.45 months and a median PFS of 8.98 months.

We also examined the effect of treatment sequencing on survival. Patients who received biologic therapy in both the first and second lines of treatment had numerically longer median OS than those who received chemotherapy only in both lines (31.74 vs. 25.54 months, respectively) or only one line of biologic therapy. The optimal order for using anti-VEGF and anti-EGFR agents in patients with wild-type *RAS* is unknown. A previous study of 490 patients found no significant difference in median OS between patients who received an anti-VEGF agent in the first line and an anti-EGFR agent in the second line, compared with the reverse sequence (31.4 vs. 31.8 months, respectively) [30]. In contrast, we observed numerically longer median OS in patients who received anti-EGFR first and anti-VEGF second (31.93 months), compared with the reverse sequence (25.41 months). Neither study can definitively describe the optimal order of biologic therapy use, although the 2017 meta-analysis by Arnold and colleagues [20] suggested that the best therapeutic strategy in patients with wild-type *RAS* with a left-sided tumor would be to start with a combination of chemotherapy plus anti-EGFR therapy followed by second-line chemotherapy plus an anti-VEGF agent. Further studies investigating this are warranted.

In patients with mutated *RAS*, longer median OS was observed when anti-VEGF treatment was used in both the first and second treatment lines than when chemotherapy alone was used in both. Interestingly, however, in the same subset of patients, there was little evidence from individual treatment lines that anti-VEGF therapy was associated with significant improvements in OS or PFS, compared with chemotherapy alone. Anti-VEGF therapy significantly improved PFS, relative to chemotherapy only, in first-line treatment, but not in the second or third lines; in addition, no OS benefit was observed in the first line. 

Regarding primary tumor location, median OS and PFS were consistently longer in patients with left-sided versus right-sided primary disease, regardless of treatment type.

In the univariate analyses, we found that younger age (≤70 years), left-sided disease, a solitary metastatic site, and the inclusion of a biologic agent in first-line therapy were all significantly associated with improved OS and PFS. Additionally, wild-type *RAS* mutational status and female sex were found to be predictive of OS but not PFS. In comparison, surgical resection—of either the primary tumor or metastases—was more strongly predictive of both OS and PFS. A favorable association between metastasectomy and prolonged OS in mCRC has been described previously [31]. It is unclear whether this effect is due to the surgery itself or due to factors associated with surgical candidacy.

In the multivariate analyses, the only interaction that was significantly associated with both OS and PFS was between age and sex, with women aged ≤70 years having favorable outcomes.

The key strength of our study lies in its depth and breadth. To the best of our knowledge, no other real-world studies of mCRC conducted since the introduction of targeted therapy describe the interactions between patient and tumor characteristics, systemic treatment patterns, and survival outcomes. We have also not identified any studies in which survival outcomes were simultaneously analyzed by treatment line, primary tumor location, *RAS* mutational status, treatment type (targeted + chemotherapy vs. chemotherapy alone), and treatment sequence. Many real-world studies of systemic treatment for mCRC have investigated outcomes with specific drugs in a single line of therapy or in a specific, defined patient population [32,33,34,35,36,37,38,39,40,41,42]. Relative to these studies, our work has a broader scope and, thus, greater potential utility, providing survival outcomes for the more comprehensive range of patients and scenarios that oncologists are likely to encounter daily. Additionally, by including centers in different autonomous regions, we reduced the impact of local and regional variations in treatment availability and practice. Therefore, our findings are both representative of and relevant to clinical practice in Spain. Finally, the large sample size ( > 2000 patients) allows a degree of confidence in the results overall and in those of the primary subgroup analyses. However, stratification by two or more variables inevitably reduces sample size, which should be considered when interpreting the findings.

In general, the findings from retrospective observational studies should be interpreted with caution because of potential confounding factors and continually evolving clinical practice. Our study only included patients treated at Spanish hospitals between 2011 and 2015. Thus, while the data may be helpful for benchmarking purposes, their applicability to other settings and healthcare systems is likely to be limited.

## 5. Conclusions

The RWD-ACROSS study provides detailed real-world data on patient and tumor characteristics, treatment patterns, and survival among patients receiving systemic treatment for mCRC in Spain. Its findings may serve as a useful benchmark for future studies of real-world outcomes in mCRC and assist practicing oncologists in selecting appropriate treatments for their patients.

## Figures and Tables

**Figure 1 cancers-15-04603-f001:**
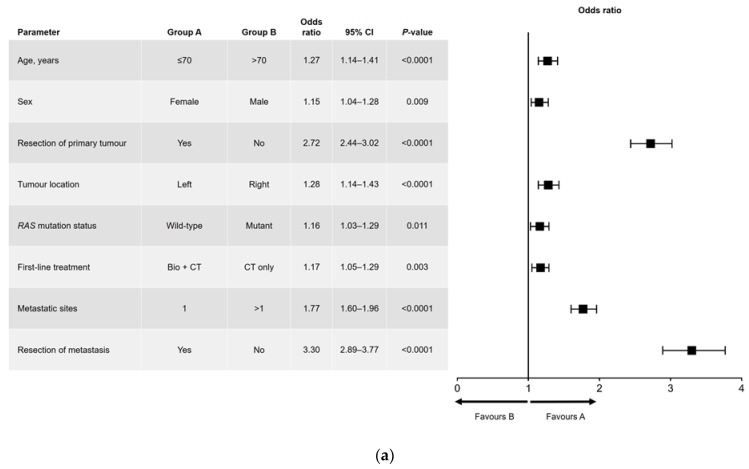
(**a**) Univariate and (**b**) multivariate analyses of overall survival. Bio, biologic therapy; CI, confidence interval; CT, chemotherapy.

**Figure 2 cancers-15-04603-f002:**
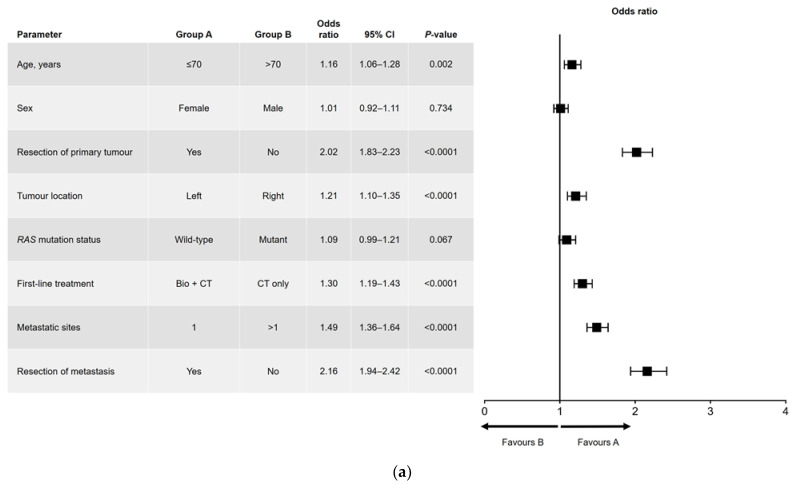
(**a**) Univariate and (**b**) multivariate analyses of progression-free survival. Bio, biologic therapy; CI, confidence interval; CT, chemotherapy.

**Table 1 cancers-15-04603-t001:** Patient demographic and tumor characteristics by line of therapy.

Parameter	Line of Therapy		
	First (n = 2002)	Second (n = 1428)	Third (n = 725)
Age, years	65.34 (20.11–89.42)	64.66 (20.11–89.42)	63.60 (20.11–86.26)
Male sex, n (%)	1256 (62.7)	890 (62.4)	431 (59.4)
Body mass index, kg/m^2^	25.59 (15.09–50.78)	–	–
Primary tumor location, n (%)			
Right	534 (26.7)	369 (25.9)	181 (25.0)
Left	1444 (72.1)	1040 (72.8)	529 (73.0)
Both	24 (1.2)	19 (1.3)	15 (2.0)
Number of metastatic sites, n (%)			
1	1185 (59.2)	–	–
2	600 (30.0)	–	–
≥3	217 (10.8)	–	–
Location of metastases, n (%)			
Patients with a single metastatic site			
Liver	754 (37.7)	–	–
Lung	164 (8.2)	–	–
Peritoneum	152 (7.6)	–	–
Lymph node	58 (2.9)	–	–
Regional	37 (1.8)	–	–
Ovary	5 (0.2)	–	–
Bone	7 (0.3)	–	–
Other	8 (0.3)	–	–
Patients with any metastatic site			
Liver + other	1404 (70.1)	–	–
Lung + other	625 (31.2)	–	–
Peritoneum + other	456 (22.7)	–	–
Lymph node + other	348 (17.4)	–	–
Resection, n (%)			
Primary tumor	1294 (64.6)	–	–
Metastasis	511 (25.5)	–	–
Mutation status, n (%)			
Known	1675 (83.7)	1261 (88.3)	680 (93.8)
*RAS* and *BRAF* wild type	821 (49.0)	604 (47.9)	342 (50.3)
*RAS* mutated	828 (49.4)	633 (50.2)	328 (48.2)
*BRAF* mutated	26 (1.6)	24 (1.9)	10 (1.5)
Unknown	327 (16.3)	167 (11.7)	45 (6.2)

Values are presented as mean (95% confidence interval) unless stated otherwise.

**Table 2 cancers-15-04603-t002:** Systemic treatments in first, second and third line.

	Line of Therapy		
	First	Second	Third
**Overall**			
Number of patients	(n = 2002)	(n = 1428)	(n = 725)
**Chemotherapy**			
Any regimen	2002 (100.0)	1376 ^1^ (100.0)	666 ^1^ (100.0)
Fluoropyrimidine only	242 (12.1)	152 (11.1)	176 (26.4)
Oxaliplatin	1316 (65.7)	375 (27.3)	160 (24.0)
Irinotecan	403 (20.1)	824 (60.1)	285 (42.8)
Oxaliplatin + irinotecan ^2^	41 (2.0)	21 (1.5)	3 (0.5)
Other	0	4 (0.3)	42 (6.3)
**Targeted therapy**			
Any targeted therapy	1130 (100.0)	806 (100.0)	395 (100.0)
Bevacizumab	702 (62.1)	402 (49.8)	172 (43.6)
Cetuximab	281 (24.9)	164 (20.4)	110 (27.8)
Panitumumab	134 (11.9)	81 (10.1)	47 (11.9)
Aflibercept	7 (0.6)	150 (18.6)	37 (9.4)
Regorafenib	6 (0.5)	8 (1.0)	28 (7.1)
Other	0	1 (0.1)	1 (0.2)
**Treatment by primary tumor location and mutation status**			
Number of patients	(n = 1626)	(n = 1219)	(n = 656)
Right-sided, wild-type *RAS*	195 (12.0)	133 (10.9)	68 (10.4)
Anti-VEGF ^3^	49 (25.2)	42 (31.5)	20 (29.4)
Anti-EGFR	74 (37.9)	46 (34.6)	26 (38.2)
Chemotherapy only	72 (36.9)	45 (33.9)	22 (32.4)
Right-sided, mutated *RAS*	224 (13.8)	171 (14.0)	92 (14.0)
Anti-VEGF ^3^	107 (47.7)	96 (56.1)	33 (35.9)
Anti-EGFR	3 (1.3)	1 (0.6)	2 (2.1)
Chemotherapy only	114 (51.0)	74 (43.3)	57 (62.0)
Left-sided, wild-type *RAS*	618 (38.0)	465 (38.1)	269 (41.0)
Anti-VEGF ^3^	146 (23.6)	146 (31.4)	66 (24.5)
Anti-EGFR	280 (45.3)	173 (37.2)	121 (45.0)
Chemotherapy only	192 (31.1)	146 (31.4)	82 (30.5)
Left-sided, mutated *RAS*	589 (36.2)	450 (36.9)	227 (34.6)
Anti-VEGF ^3^	308 (52.3)	219 (48.7)	98 (43.1)
Anti-EGFR	19 (3.2)	8 (1.8)	4 (1.8)
Chemotherapy only	262 (44.5)	223 (49.5)	125 (55.1)

All values are presented as n (%). ^1^ Patients who received chemotherapy as second- and third-line treatment excluded those who died on first-line treatment and those who underwent surgery for metastasis removal. ^2^ FOLFIRINOX. ^3^ Anti-VEGF therapy includes treatment with regorafenib. EGFR, endothelial growth factor receptor; VEGF, vascular endothelial growth factor.

**Table 3 cancers-15-04603-t003:** Overall survival and progression-free survival for the study group as a whole, and by primary tumor location and *RAS* mutation status, following first-line treatment.

Outcome	All Patients (n = 2002)	Primary Tumor Location (n = 1978) ^1^			*RAS* Mutation Status (n = 1649) ^2^		
		Left (n = 1444)	Right (n = 534)	*p*-Value	Wild-Type (n = 821)	Mutant (n = 828)	*p*-Value
OS, months	26.72 (25.37–28.07)	28.85 (27.36–30.34)	21.04 (18.87–23.22)	<0.0001	29.04 (26.98–31.11)	27.54 (25.57–29.50)	0.032
PFS, months	10.72 (10.24–11.19)	11.24 (10.63–11.85)	9.31 (8.60–10.01)	<0.0001	11.24 (10.53–11.95)	10.78 (10.04–11.53)	0.096

Values are presented as median (95% confidence interval) unless stated otherwise. ^1^ An additional 24 patients had primary tumors in both the left and right colon at study entry, and are excluded. ^2^ Mutation status was unknown in 327 patients and an additional 26 had mutant *BRAF*. CI, confidence interval; OS, overall survival; PFS, progression-free survival. *p*-value < 0.05 was considered statistically significant.

**Table 4 cancers-15-04603-t004:** Overall survival in patients receiving first-line systemic treatment (n = 1675), according to location of primary tumor, *RAS* mutation status, and type of systemic treatment.

Primary Tumor Location	*RAS* Mutation Status	Any Treatment	Systemic Treatment			*p*-Value
			Anti-VEGF	Anti-EGFR	Chemotherapy Only	
Unselected	Unselected	(n = 1675)	(n = 715)	(n = 415)	(n = 872)	0.002
		28.03 (26.65–29.41)	26.75 (24.67–28.83)	31.21 (27.78–34.63)	24.45 (22.51–26.40)	
	Wild type	(n = 821)	(n = 197)	(n = 358)	(n = 266)	0.963
		29.04 (26.98–31.11)	26.62 (23.01–30.23)	30.06 (26.41–33.71)	29.14 (25.99–32.30)	
	Mutant	(n = 828)	(n = 421)	(n = 22)	(n = 385)	0.140
		27.54 (25.57–29.50)	28.85 (26.48–31.21)	32.19 (22.92–41–47)	25.34 (22.77–27.91)	
	*BRAF* mutant	(n = 26)	–	–	–	–
		23.18 (0.00–50.58)	–	–	–	
Right	Unselected	(n = 534)	(n = 185)	(n = 85)	(n = 264)	0.526
		21.04 (18.71–23.22)	22.13 (17.85–26.40)	21.63 (16.49–26.78)	19.37 (16.18–22.56)	
Left	Unselected	(n = 1444)	(n = 522)	(n = 325)	(n = 597)	0.003
		28.85 (27.36–30.34)	28.23 (25.68–30.77)	34.29 (31.04–37.54)	26.59 (24.23–28.94)	
Right	Wild type	(n = 195)	(n = 49)	(n = 74)	(n = 72)	0.201
		21.08 (17.45–24.70)	18.62 (15.04–22.20)	19.31 (12.83–25.78)	24.62 (19.99–29.25)	
	Mutant	(n = 224)	(n = 107)	(n = 3)	(n = 114)	0.907
		23.86 (20.15–27.58)	23.47 (17.64–29.31)	32.19 (–)	23.86 (17.84–29.89)	
Left	Wild type	(n = 618)	(n = 146)	(n = 280)	(n = 192)	0.583
		31.50 (28.93–34.08)	28.06 (23.68–32.44)	33.86 (30.43–37.30)	30.26 (26.75–33.76)	
	Mutant	(n = 589)	(n = 308)	(n = 19)	(n = 262)	0.182
		28.26 (25.98–30.54)	30.45 (27.56–33.35)	32.65 (15.83–49.48)	26.13 (22.80–29.45)	

Values are presented as median (95% confidence interval) months unless stated otherwise. *BRAF*, B-Raf proto-oncogene serine/threonine kinase; EGFR, endothelial growth factor receptor; VEGF, vascular endothelial growth factor. *p*-value < 0.05 was considered statistically significant.

**Table 5 cancers-15-04603-t005:** Progression-free survival in patients receiving first-, second-, or third-line systemic treatment, according to location of primary tumor, *RAS* mutation status, and type of systemic treatment.

Primary Tumor Location	*RAS* Mutation Status	Any Treatment	Systemic Treatment			*p*-Value
			Anti-VEGF	Anti-EGFR	Chemotherapy Only	
First line (n = 2002)						
Unselected	Unselected	(n = 1675)	(n = 715)	(n = 415)	(n = 872)	<0.001
		11.01 (10.54–11.48)	12.13 (11.36–12.89)	12.00 (11.01–12.99)	8.98 (8.42–9.54)	
	Wild type	(n = 821)	(n = 197)	(n = 358)	(n = 266)	0.055
		11.24 (10.53–11.95)	12.23 (10.75–14.31)	11.76 (10.44–12.97)	10.25 (8.12–12.40)	
	Mutant	(n = 828)	(n = 421)	(n = 22)	(n = 385)	<0.0001
		10.78 (10.04–11.53)	12.36 (11.33–13.38)	12.03 (8.34–15.72)	8.98 (8.37–9.59)	
	*BRAF* mutant	(n = 26)	–	–	–	–
		10.19 (3.88–16.50)	–	–	–	
Right	Unselected	(n = 534)	(n = 185)	(n = 85)	(n = 264)	0.002
		9.31 (8.60–10.01)	11.44 (9.93–12.95)	10.75 (8.21–13.28)	7.77 (6.81–8.73)	
Left	Unselected	(n = 1444)	(n = 522)	(n = 325)	(n = 597)	<0.0001
		11.24 (10.63–11.85)	12.39 (11.44–13.34)	13.21 (11.43–14.99)	9.50 (8.79–10.21)	
Right	Wild type	(n = 195)	(n = 49)	(n = 74)	(n = 72)	0.025
		9.90 (8.51–11.29)	11.24 (9.22–13.26)	8.78 (6.27–11.30)	7.83 (4.67–10.99)	
	Mutant	(n = 224)	(n = 107)	(n = 3)	(n = 114)	0.056
		10.03 (8.60–11.45)	12.09 (9.70–14.49)	16.45 (7.59–25.32)	8.85 (8.03–9.67)	
Left	Wild type	(n = 618)	(n = 146)	(n = 280)	(n = 192)	0.011
		11.70 (10.72–12.68)	12.39 (10.57–14.21)	13.14 (11.40–14.89)	9.83 (8.61–11.05)	
	Mutant	(n = 589)	(n = 308)	(n = 19)	(n = 262)	0.001
		10.98 (10.11–11.85)	12.36 (11.16–13.55)	12.03 (10.77–13.29)	9.34 (8.45–10.23)	
Second line (n = 1427)						
Unselected	Unselected	(n = 1427)	(n = 560)	(n = 245)	(n = 622)	0.001
		7.63 (7.25–8.02)	8.26 (7.55–8.97)	8.82 (7.59–10.04)	6.68 (6.06–7.31)	
	Wild-type	(n = 604)	(n = 190)	(n = 219)	(n = 195)	0.068
		8.29 (7.68–8.90)	8.55 (7.66–9.44)	9.18 (7.57–10.78)	7.31 (6.38–8.24)	
	Mutant	(n = 632)	(n = 321)	(n = 9)	(n = 302)	0.098
		7.34 (6.80–7.88)	7.86 (6.87–8.86)	8.39 (4.08–12.70)	6.98 (6.11–7.85)	
	*BRAF* mutant	(n = 24)	–	–	–	–
		5.47 (3.82–7.12)	–	–	–	
Right	Unselected	(n = 369)	(n = 159)	(n = 52)	(n = 158)	0.006
		7.08 (6.45–7.71)	7.34 (6.30–8.38)	7.83 (6.07–9.60)	5.80 (4.34–7.26)	
Left	Unselected	(n = 1040)	(n = 393)	(n = 192)	(n = 455)	0.012
		8.00 (7.48–8.52)	8.91 (8.10–9.73)	8.91 (7.52–10.30)	7.08 (6.35–7.80)	
Right	Wild-type	(n = 133)	(n = 42)	(n = 46)	(n = 45)	0.085
		7.57 (6.68–8.46)	7.57 (5.83–9.30)	7.83 (5.12–10.54)	6.68 (4.04–9.33)	
	Mutant	(n = 171)	(n = 96)	(n = 1)	(n = 74)	0.090
		7.08 (6.04–8.12)	7.24 (5.85–8.64)	8.39 (–)	6.42 (3.79–9.05)	
Left	Wild-type	(n = 465)	(n = 146)	(n = 173)	(n = 146)	0.256
		8.75 (8.06–9.44)	9.01 (7.85–10.17)	9.57 (7.85–11.29)	7.44 (6.14–8.74)	
	Mutant	(n = 449)	(n = 219)	(n = 8)	(n = 223)	0.316
		7.50 (6.81–8.20)	8.26 (7.03–9.49)	6.91 (1.27–12.56)	7.08 (6.19–7.97)	
Third line (n = 723)						
Unselected	Unselected	(n = 723)	(n = 237)	(n = 156)	(n = 330)	0.037
		6.16 (5.65–6.67)	6.36 (5.60–7.12)	7.18 (6.45–7.90)	5.21 (4.40–6.02)	
	Wild-type	(n = 341)	(n = 88)	(n = 148)	(n = 105)	0.468
		6.85 (6.16–7.54)	6.42 (4.70–8.14)	7.24 (6.44–8.05)	6.52 (4.72–8.32)	
	Mutant	(n = 327)	(n = 134)	(n = 6)	(n = 187)	0.005
		5.37 (4.68–6.07)	6.36 (5.67–7.05)	2.98 (0.00–6.80)	4.82 (4.35–5.28)	
	*BRAF* mutant	(n = 10)	–	–	–	–
		6.29 (4.31–8.27)	–	–	–	
Right	Unselected	(n = 181)	(n = 55)	(n = 28)	(n = 98)	0.720
		4.95 (4.33–5.56)	4.98 (4.03–5.93)	5.01 (2.84–7.18)	4.59 (3.85–5.32)	
Left	Unselected	(n = 529)	(n = 176)	(n = 127)	(n = 226)	0.088
		6.62 (6.03–7.21)	6.62 (5.69–7.54)	7.27 (6.60–7.95)	5.67 (4.66–6.67)	
Right	Wild-type	(n = 68)	(n = 20)	(n = 26)	(n = 22)	0.996
		5.08 (3.19–6.97)	5.08 (2.35–7.81)	5.27 (1.22–9.33)	4.91 (1.71–8.12)	
	Mutant	(n = 92)	(n = 33)	(n = 2)	(n = 57)	0.129
		4.52 (3.81–5.23)	4.78 (3.71–5.85)	1.83 (–)	4.29 (3.53–5.05)	
Left	Wild-type	(n = 269)	(n = 66)	(n = 121)	(n = 82)	0.455
		7.08 (6.39–7.76)	6.85 (5.05–8.64)	7.44 (6.59–8.29)	6.65 (4.98–8.32)	
	Mutant	(n = 226)	(n = 98)	(n = 4)	(n = 124)	0.107
		6.13 (5.34–6.91)	6.62 (5.47–7.76)	2.98 (0.00–9.47)	5.21 (4.09–6.33)	

Values are presented as median (95% confidence interval) months unless stated otherwise. *BRAF*, B-Raf proto-oncogene serine/threonine kinase; EGFR, endothelial growth factor receptor; VEGF, vascular endothelial growth factor. *p*-value < 0.05 was considered statistically significant.

## Data Availability

All data generated/analyzed in the current study are available from the corresponding author upon reasonable request.

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
