# Peer review of "Real-World Outcomes in Patients with Metastatic Colorectal Cancer in Spain: The RWD-ACROSS Study"

_cancers, 2023, doi:10.3390/cancers15184603_

Round 1

Reviewer 1 Report

Comments for cancers-2492494

Carles Pericay, et al.  Real-world outcomes in patients with metastatic colorectal cancer in 

Spain: the RWD-ACROSS study. 

The authors analyzed the association between treatments and survival of patients with mCRC by primary tumor location and mutation status. This study is important and interesting in clinical oncology. As the authors wrote, this paper may serve as a useful benchmark for future studies and assist practicing oncologists in selecting appropriate treatments for their patients. 

Author Response

We thank the reviewer for their positive comments about our manuscript.

Reviewer 2 Report

The Manuscript cancers-2492494-peer-review-v1

Manuscript cancers-2492494-peer-review-v1 describes multicenter outcomes for the treatment of metastatic colorectal cancer in Spain.

The manuscript is well-written and within the scope of cancers. It reveals important information that will be of interest to the readers of the journal. This information could be utilized by other investigators to support the generation of reliable guidelines in the treatment of mCRC.

Author Response

We are grateful to the reviewer for their encouraging words about our manuscript.

Reviewer 3 Report

This is a multi-institutional retrospective study investigating the real-world clinical practice for patients with metastatic colorectal carcinoma (mCRC) and their long-term outcome in Spain. The authors concluded that wild-type RAS or left-sided mCRC were predictive of longer survival in Spain. However, the following concerns need to be addressed:

1.        During the study period (2011-2015), it would not be general yet to evaluate all RAS status (i.e. KRAS, NRAS, and HRAS) because the importance of other RAS family gene than KRAS gene in mCRC patients was first reported in 2010 (Lancet Oncol 2010; 11: 753–62). The proportion of RAS mutation shown in this article should be underestimated.

2.        While the authors emphasized their data as “real-world” in Spain, there is a selection bias in this study cohort. It is difficult to understand why they enrolled the same number of patients from each center without the consideration for its sample size.

3.        The authors included mCRC patients who received ≥1 cycle of chemotherapy for metastatic disease. It means that patients who were cured by only surgical therapy were excluded as well as patients who chose best supportive care, while surgical resection is widely known as the gold standard approach for resectable mCRC, especially in liver and/or lung. It is another selection bias.

4.        As commented in #3, the evaluation regarding the resectability of mCRC in liver and/or lung is essential for the decision of the current treatment approach for mCRC. However, no information for deciding the resectability of mCRC, such as performance status, the number of tumor, and tumor location (unilober or bilobar) was shown in this study.

5.        According to Table 1, 25.5% of patients underwent the resection of metastatic site, and the authors should state how they defined progression-free survival in such patients. If it was similar to recurrence-free survival (RFS), it is not appropriate to treat them equally. Additionally, they should show the data how many patients underwent the resection for the recurrent disease.

Author Response

  1. During the study period (2011-2015), it would not be general yet to evaluate all RAS status (i.e. KRAS, NRAS, and HRAS) because the importance of other RAS family gene than KRAS gene in mCRC patients was first reported in 2010 (Lancet Oncol 2010; 11: 753–62). The proportion of RAS mutation shown in this article should be underestimated.

Authors’ response: The inclusion period for patients in our study was January 2011 to December 2015. During this time, there was a progressive increase in the clinical determination of KRAS, NRAS, and BRAF mutations in patients diagnosed with metastatic colorectal cancer. As described in our study, the frequency of mutation status determination increased as patients lived longer and underwent second- and third-line treatments (the study was closed for inclusion at the end of 2019). We believe that having RAS and BRAF mutation status known in 83.7% of patients during first-line treatment from 2011 (see Table 1 in the manuscript) represents a success for clinical practice in Spain. Furthermore, up to 93.8% of patients had RAS and BRAF mutation status determined by the time they reached third-line treatment, which means that, by 2015 (the end of our study period), mutation status assessment had become a very common practice in Spanish hospitals.

Our series included all patients who had received at least one cycle of chemotherapy, including those who received monotherapy with 5-fluorouracil or capecitabine at reduced doses if their ECOG score was 2 or 3. The hospitals that provided the cases for our study are university hospitals that deliver specialist medical training, i.e., treatment facilities that are likely to provide high-quality treatment and follow-up of patients. Therefore, we do not believe that the values for mutation determination in our study are underestimated.

  1. While the authors emphasized their data as “real-world” in Spain, there is a selection bias in this study cohort. It is difficult to understand why they enrolled the same number of patients from each center without the consideration for its sample size.

Authors’ response: We appreciate the reviewer’s comment and we agree that the description of the study methodology was not clear. To clarify, data from all patients treated with at least one cycle of chemotherapy at the 10 study centers were collected. The centers included all patients from their hospital registry who met the inclusion criteria, up to a maximum of 250 patients (although one hospital was allowed to include 300 patients because several other hospitals had included fewer than 250 patients). This has been clarified in the ‘Inclusion and exclusion criteria’ section of the Methods (paragraph 1). The centers were distributed throughout Spain and they provided data from patients treated with standard clinical protocols; thus, we consider that our study collected real-world data. Further, it is the largest such study from Spain of patients with metastatic adenocarcinoma of the colon.

  1. The authors included mCRC patients who received ≥1 cycle of chemotherapy for metastatic disease. It means that patients who were cured by only surgical therapy were excluded as well as patients who chose best supportive care, while surgical resection is widely known as the gold standard approach for resectable mCRC, especially in liver and/or lung. It is another selection bias.

Authors’ response: The aim of our study was to assess the effect of chemotherapy in patients with metastatic colorectal cancer. The administration of chemotherapy is performed according to established protocols that are widely disseminated throughout the oncology community, which facilitates collection of data from a relatively homogenous sample. In addition, patients who have received chemotherapy are easier to identify via pharmacy and day hospital registries than those who have received other forms of treatment. For example, complete data from patients who received best supportive care alone is difficult to obtain because these patients may be admitted to palliative care facilities or monitored by home-care teams and their data are not readily available. Further, very few patients receive only surgery for metastatic disease, with most patients receiving a course of adjuvant chemotherapy, and there is apparent variability in surgical criteria, which makes surgical patients a very heterogeneous group (Folprecht G, et al. Ann Oncol. 2014 May; 25(5): 1018-25). For these reasons, we considered that the most appropriate methodology to obtain a homogenous sample was to select patients who had received chemotherapy and were included in oncology day hospital registers or pharmacy registers.

  1. As commented in #3, the evaluation regarding the resectability of mCRC in liver and/or lung is essential for the decision of the current treatment approach for mCRC. However, no information for deciding the resectability of mCRC, such as performance status, the number of tumor, and tumor location (unilober or bilobar) was shown in this study.

Authors’ response: RWD-ACROSS was a retrospective study, based on data from medical records. Performance status data are only valid in prospective studies and, therefore, were not collected in our study. Further, being a retrospective study, no data on the resectability of the patients’ tumors were available, only data on whether or not there had been resection. Data were also not collected on whether the resection was of a single lesion or multiple lesions within the same organ.

  1. According to Table 1, 25.5% of patients underwent the resection of metastatic site, and the authors should state how they defined progression-free survival in such patients. If it was similar to recurrence-free survival (RFS), it is not appropriate to treat them equally. Additionally, they should show the data how many patients underwent the resection for the recurrent disease.

Authors’ response: We considered surgical resection of metastases (with or without associated chemotherapy) as a first-line treatment and, therefore, we used progression-free survival (not recurrence-free survival) in these patients. Given that these patients had stage IV cancer, ‘progression’ is the most appropriate terminology to use (median progression-free survival was 18.82 [95% confidence interval 16.94–20.70] months in patients who underwent hepatectomy [data not shown]). Unfortunately, we currently do not have detailed analyses of resection for recurrent disease, and so we could not add these data.

Round 2

Reviewer 3 Report

The authors answered to the reviewers' comments well and the article was well revised.